

# Estimation and prediction of water conservation in the upper reaches of the Hanjiang River Basin based on InVEST-PLUS model

Pengtao Niu[1,2], Zhan Wang[2], Jing Wang[1], Yi Cao[3,4] and Peihao Peng[1]

[1] College of Geography and Planning, Chengdu University of Technology, Chengdu, Sichuan, China
[2] School of Surveying Engineering and Environment, Henan Polytechnic Institute, Nanyang, Henan, China
[3] Sinopec Northwest China Petroleum Bureau, Urumqi, Xinjiang, China
[4] School of Sciences and Engineering, Hohai University, Nanjing, Jiangsu, China

Corresponding author
Peihao Peng, pengpeihao@cdut.edu.cn

## ABSTRACT

With the gradual prominence of global water shortage and other problems, evaluating and predicting the impact of land use change on regional water conservation function is of great reference significance for carrying out national spatial planning and environmental protection, and realizing land intelligent management. We first analyzed 8,416 remote sensing images in the upper reaches of the Hanjiang River Basin (URHRB) by GEE platform and obtained the land use and land cover (LULC) results of fours periods. Through our field investigation, the accuracy of remote sensing image interpretation is obviously higher than that of other comprehensive LULC classification results. Then, through the coupling of InVEST-PLUS model, not only the results of URHRB water conservation from 1990 to 2020 were calculated and the accuracy was assessed, but also the LULC results and water conservation of URHRB under different development scenarios in 2030 were predicted. The results showed as follows: From 1990 to 2020, the forest area of URHRB increased by 7152.23 km$^2$, while the area of cropland, shrub and grassland decreased by 3220.35 km$^2$, 1414.72 km$^2$ and 3385.39 km$^2$, respectively. The InVEST model reliably quantifies the water yield and water conservation of URHRB. In the past 30 years, the total amount of water-saving in China has shown a trend of increasing first and then decreasing. From the perspective of vegetation types, URHRB forest land is the main body of water conservation, with an average annual water conservation depth of 653.87 mm and an average annual water conservation of 472.10×108 m$^3$. Under the ecological protection scenario of the URHRB in 2030, the maximum water conservation in the basin is 574.92×108 m$^3$, but compared with the water conservation in 2010, there is still a gap of 116.28×108 m$^3$. Therefore, through the visualization analysis of the LULC changes of URHRB and water conservation function, it is found that the land and resources department should pay attention to the LULC changes of water sources and adjust the territorial spatial planning in time to cope with the huge water conservation gap in the future.

# INTRODUCTION

Water resources are vital for human survival (*Zhou et al., 2015*). The main sources of water re-sources available to human are atmospheric precipitation (*Yang, Hou & Cao, 2023*), surface water and groundwater (*Francesconi et al., 2016*; *Shi et al., 2023*). The production and retention of these water resources are closely linked to the ecosystem's water conservation function (*Li, 2022*; *Ma et al., 2022*; *Zeng et al., 2022*). Water conservation function entails conserving soil and water, regulating surface runoff, and ultimately altering the regional water supply pattern by capturing and storing rainwater *via* the forest canopy, litter layer, and soil layer (*Heffernan & Strimbu, 2021*; *Kezik & Hacisalihoglu, 2022*; *Wu & Gu, 2020*). In the process of water conservation, spatial heterogeneity exists between land use and the surface water and groundwater cycles (*Khoroshev & Emelyanova, 2024*). Inappropriate vegetation restoration and human activity interference can impact regional water conservation function. In recent years, amidst global warming (*Tollefson, 2019*), there has been a general decline in regional water conservation capacity. This has significantly impacted regional sustainable development and has garnered considerable attention from many countries. As water scarcity jeopardizes water supply safety, ecological integrity, and food security, it serves as a pivotal factor constraining high-quality economic and social development. The Middle Route of China's South-to-North Water Diversion project (SNWD) has demonstrated that scientific management of water transfer is an important way to alleviate the shortage of regional water resources in China (*Liu & Guo, 2020*; *Sun et al., 2023*; *Zhu et al., 2021*). The upper reaches of the Hanjiang River Basin (URHRB) are the core water source area of the Middle Route of the SNWD. To this end, in recent years, driven by the imperatives of ecological preservation and high-quality development, China has enhanced the equilibrium between territorial space and water resources through the implementation of cross-basin and cross-regional water transfer initiatives (*Niu et al., 2022*). Concurrently, it has undertaken ecological environment management within each water catchment area, systematically restoring extensive ecological land, and enhancing the water conservation capacity of each catchment area. These efforts provide a sturdy groundwork for elevating regional ecological governance standards (*Wang & Ma, 1999*). Hence, it is imperative to investigate the impact of dynamic land use and land cover (LULC) changes on water conservation function in the URHRB within the framework of environmental governance.

Currently, numerous scholars employ various models such as the MIKE SHE model (*Jaber & Shukla, 2012*; *Reszler & Fank, 2016*), SWAT (*Arnold et al., 2012*; *Moriasi et al., 2012*; *Rodrigues, 2017*; *Zhang, Srinivasan & Bosch, 2009*) and InVEST model (*Allen et al., 2005*; *Donohue, Roderick & McVicar, 2012*; *Wang et al., 2023a*) to conduct detailed simulations and evaluations of regional water conservation function. Among these, the InVEST model, jointly developed by Stanford University, CAS (Chinese Academy of Sciences), WWF (World Wide Fund for Nature) and other internationally renowned scientific research institutions, features flexible parameter adjustment and visualization of evaluation results (*Hamel & Guswa, 2015*). The InVEST model has found extensive application in ecological environment assessments worldwide, such as South China

Monsoon basin (*Yang et al., 2020*), the Shule River basin (*Wei et al., 2021*), Qinling Mountains (*Li et al., 2021b*), Yellow River source area (*Ding et al., 2021*) and other places (*Hamel & Guswa, 2015*; *Zhao et al., 2022*).

As land use and land cover (LULC) change emerges as a crucial limiting factor for water conservation, necessitating accurate calculation and evaluation of such changes as a primary step. As early as the early 19th century, European scholars initiated studies on land use with a focus on achieving sustainable development (*Reenberg, Nielsen & Rasmussen, 1998*). Land constitutes a crucial component of ecosystem ser-vices, and alterations in LULC types directly influence regional ecological environments, thereby impacting the stability of regional ecosystems (*Huo et al., 2022*; *Montanarella & Panagos, 2021*; *Sierra-Soler et al., 2016*). Spatial–temporal dynamic analysis of LULC necessitates monitoring methods encompassing extensive spatial coverage, long-term time series, and high precision. The continuity in producing LULC products over large areas poses a challenge of high cost to traditional remote sensing monitoring and data processing methods. The advent of satellite remote sensing cloud storage and cloud computing platforms, exemplified by the GEE cloud platform, has enabled large-scale production of LULC. Methods for predicting LULC changes include the CA (cellular automata) model, the future land use simulation (FLUS) model, the patch-generating land use simulation (PLUS) model and others. Furthermore, the IMAGE, LUSs, and CLUMondo models are also under research (*Kombate et al., 2022*; *Sertel et al., 2022*; *Zhu et al., 2020*). The PLUS model is a novel land use prediction model based on the FLUS model (*Liang et al., 2021*). PLUS model addresses deficiencies in current land use prediction models concerning various patch scales. The PLUS model can integrally incorporate the rule framework based on the land expansion analysis strategy (LEAS) with the CA model to achieve de-tailed simulations of future LULC changes (*Liang et al., 2018*; *Liu et al., 2017*). Several scholars (*Mo et al., 2021*) conducted numerical simulations of land use status in the headwaters of the Dongjiang River using the PLUS-INVEST model. *Liu, Zhang & Lin (2023)* analyzed the carbon storage function of the Junge Banner using the InVEST and FLUS models, and predicted its developmental trajectory. *Zhou et al. (2024)* calculated and assessed the water conservation capacity of Northwest China from 1990 to 2020 by using the InVEST model and PLUS model, and simulated the water conservation capacity of this region under three development scenarios combined with climate factors. While some scholars have investigated and deliberated upon the ecological development scenario of the headwater catchment of the Middle Route of the SNWD (*Hu et al., 2019*; *Ren, Chen & Liu, 2007*), there is limited research on the quantitative evaluation and simulation prediction of LULC change and water conservation function within URHRB. Most of the above studies used existing large-scale LULC classification products, but most of these products have not been verified by regional inspection, and the LULC classification type may not be applicable in the URHRB. To this end, through field verification, we used the GEE platform to carry out accurate LULC classification, calculated the water conservation results of the URHRB region from 1990 to 2020, distinguished the water conservation contributions of different land use types, and simulated the LULC results of the URHRB region in 2030 under four development scenarios using the PLUS model. Finally, based on the CMIP6 climate model, the temporal and spatial dynamic

changes of water conservation under different development scenarios were predicted, and the response rules of the URHRB water conservation function to different development scenarios were analyzed. Our research will provide a basis for URHRB's water resources management and water ecological protection.

# MATERIALS & METHODS

## Overview of the study area

URHRB is the Headwater Catchment area of the Middle Route of the SNWD, located at the intersection of Gansu, Shaanxi, Chongqing, Hubei, Sichuan and Henan provinces, with Qinling Mountain region in the north and Daba Mountain region in the south, covering an area of 92,290 $km^2$. It is the water source area of Danjiangkou Reservoir, the largest artificial fresh water lake in Asia, the specific situation is shown in Fig. 1. The landform types in this area are mainly subalpine, middle mountain, low mountain, wide valley basin, karst landform and mountain ancient glacier landform. The river in URHRB belongs to the Hanjiang River system of the Yangtze River basin and is also the first tributary of the Yangtze River (*Wu et al., 2023*; *Zhang et al., 2023*). Due to the suitable climatic conditions, conducive to agricultural production.

## Research methods
### Research route

To investigate the response process of water conservation function to LULC changes in the URHRB and to predict its future state (*Xie et al., 2019*), this study initially utilized the Google Earth Engine platform to generate LULC data (30 m) for the catchment spanning 1990, 2000, 2010, and 2020. Subsequently, the study analyzed the patial-temporal variation characteristics of LULC changes in the catchment. On the basis of referring to the literature of *Hu, He & Chen (2023)* and *Lu et al. (2022)*, we comprehensively consider the current development situation of URHRB and the future socio-economic development plan (*Nie et al., 2022*), and used the PLUS model to simulate the LULC status in 2030 under four scenarios: natural development (NAS), urban development (UDS), agricultural development (ADS) and ecological protection (EPS). Subsequently, the INVEST model and mathematical formulae were utilized to calculate water conservation within URHRB spanning 1990 to 2020, and to analyze the response pattern of water conservation function to land use changes in URHRB. Ultimately, the study forecasted differences in water conservation function across the four planning and development scenarios of URHRB for 2030, aiming to offer theoretical and empirical support for territorial spatial planning, ecological restoration, and environmental governance within URHRB. The technical route and methodology diagram of this study are illustrated in Fig. 2.

## Land use classification of remote sensing image based on GEE and deep learning

In order to screen out the LULC classification system suitable for HUBRU and ensure the classification accuracy of LULC. According to the purpose of the research, we divided the LULC types into forest, grassland, bare land, cropland, shrub, construction land and water,

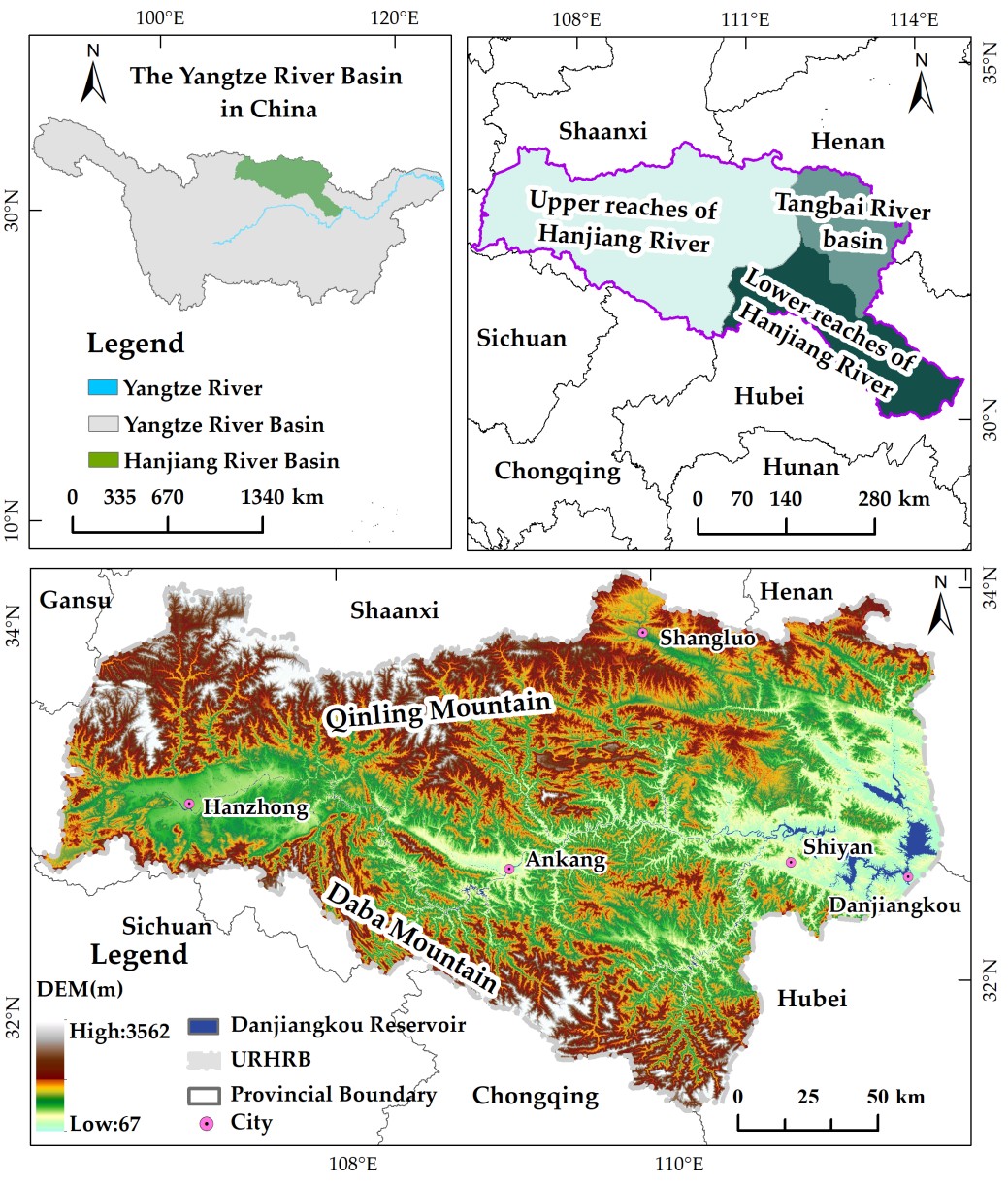

**Figure 1** Location map of the study area (URHRB). Map Source: Tianditu (http://www.tianditu.gov.cn).

with a spatial resolution of 30 m for the classification outcomes. Initially, interpretive markers were established based on the texture, spectral, and topographic characteristics of the seven types of land cover. Subsequently, the *ee.Algorithms.Landsat.simpleComposite ()* algorithm is used to synthesize the annual minimum cloud cover image data in GEE platform (*Amani et al., 2020*; *Dai, Liu & Liu, 2020*). Then, 8,416 remote sensing image discriminant samples were uniformly established in URHRB using visual interpretation methods, among which 1,935, 2,028, 2,194 and 2,359 samples allocated for 1990, 2000, 2010, and 2020, respectively. Through field investigations, 907 field observation samples were
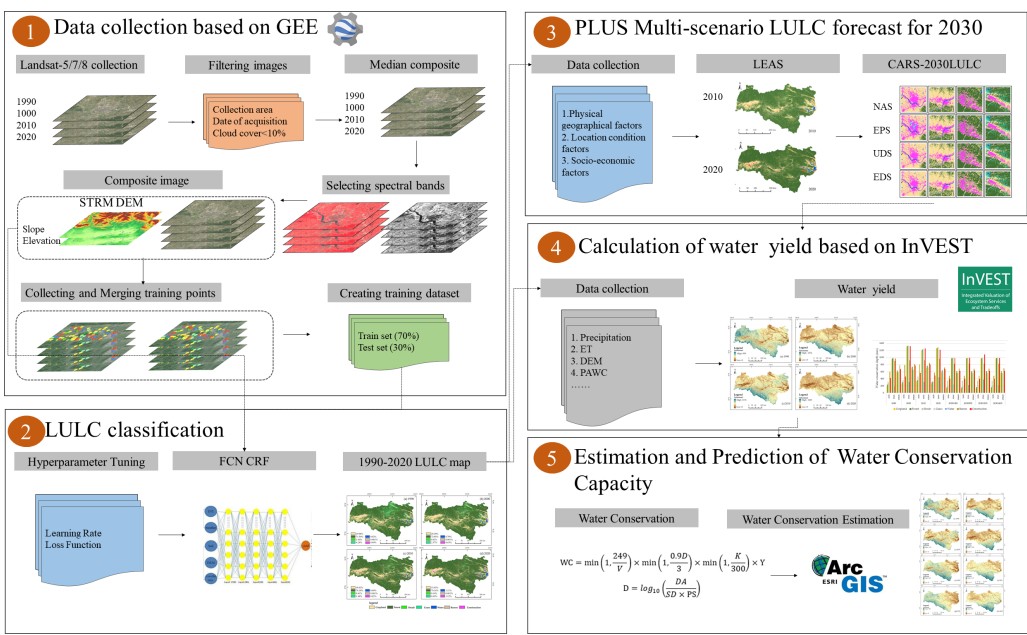

**Figure 2  Technical route and research methodology.** Map Source: Tianditu (http://www.tianditu.gov. cn).

recorded and collected. Google Earth Pro high-precision historical images were utilized to generate a total of 1,084 test samples, comprising 287 forested areas, 240 croplands, 267 shrubs, 64 grasslands, 98 construction lands, 47 bare lands, and 81 water bodies. The aforementioned trained classification model was employed to conduct inference classification and generate classification results (*Ma, Huang & Chai, 2021*). The overall classification accuracy of remote sensing image interpretation for 2020, 2010, 2000, and 1990 was 93.83%, 92.44%, 91.76%, and 89.88%, respectively. The Kappa coefficients were 0.89, 0.88, 0.86, and 0.82, respectively, indicating that the interpretation and classification results of remote sensing images met the research requirements.

## The LULC forecast for 2030 is conducted based on the PLUS model

PLUS model comprises two modules: the LEAS (Land Expansion Analysis Strategy) and the CARS (Cellular Automata based on multiple random seeds) (*Liang et al., 2021*). The LEAS module employs the Random Forest algorithm to sample these segments, and computes the probabilities of various land uses developing as well as the contributions of these driving factors. The CARS module is a cellular automata (CA) based on multiple random seeds, which can flexibly deal with multiple types of land use patch changes (*Wang et al., 2020*; *Yu, Wu & Wang, 2023*).

This study mainly adjusts the land use transfer probability matrix to create the land use demand under the four planning development scenarios in URHRB of 2030. The land use demand in the NAS scenario is calculated based on the land use expansion from 2010 to 2020. The UDS scenario is mainly based on the development planning of major local cities, and the construction land area is not less than the existing area in 2020, and

**Table 1  The parameter of neighborhood weight.**

| Annual / 2030 forecast scenarios | The parameter of neighborhood weigh | | | | | | |
|---|---|---|---|---|---|---|---|
| | Cropland | Forest | Shrub | Grass | Water | Barren | Construction |
| 2030_NAS | 0.01 | 1 | 0.41 | 0.30 | 0.52 | 0.45 | 0.51 |
| 2030_EPS | 0.01 | 1 | 0.58 | 0.64 | 0.58 | 0.56 | 0.55 |
| 2030_UDS | 0.01 | 0.43 | 0.45 | 0.41 | 0.45 | 0.46 | 1 |
| 2030_ADS | 1 | 0.01 | 0.49 | 0.48 | 0.48 | 0.49 | 0.48 |

the land use types such as forest, shrubs, and cropland can be appropriately transferred to the construction land. The ADS scenario aims to simulate the impact of cropland protection policies and reduce the transfer of cropland to other land types. Land types such as forest, grassland and construction land can be appropriately transferred to cropland. The cropland area in the simulation results is no less than the cropland holding in 2020, and the construction land and other barren land are no more than the existing land area in 2020. In the EPS scenario, ecological land such as forest, shrub, grassland and water area should be protected, the area of cropland and construction land should not exceed the existing area, and the probability of transferring other land types other than water areas to forest should be increased. At the same time, various nature reserves within URHRB should be set as restricted conversion areas. The parameter settings of land use type transfer matrix and neighborhood weight under four scenarios in the PLUS model are shown in Tables 1 and 2, respectively.

This study initially simulated the LULC outcomes for URHRB in 2020 and subsequently assessed the accuracy of these simulations using the "Plus-Validation" module, and the Kappa coefficient was 0.79, and the overall accuracy reached 0.93. Thus it is evident that the PLUS model effectively enhances the prediction and simulation of land use. During the experiment, the LEAS module initially employed a random sampling strategy to extract 1% of the land use expansion in 2020 compared to 2010 as the training sample. It set the number of hidden layers in the training model to 20, and utilized a neural network algorithm to estimate the development probabilities for various land types under the natural progression of the catchment by 2030 (*Wang et al., 2023b*). Influence factors from neighboring areas under natural development scenarios were inputted into the CARS module, which then predicted the LULC for URHRB in 2030. Finally, various land use type transfer matrices and development weights were adjusted based on historical land use changes and regional spatial development plans, resulting in LULC predictions for three additional scenarios.

## The InVEST model computes water yield and water conservation
### Calculation of water conservation

The model's water yield module was proposed by Budyko, does not distinctly differentiate between surface water, groundwater, and base flow,it finally presenting the calculation results in a graphical way (*Zhou et al., 2015*). In other words, the process involves calculating the precipitation for each grid cell and subtracting evapotranspiration, which encompasses

**Table 2 Land use type transfer matrix under four scenarios of 2030.**

| Annual / 2030 forecast scenarios | | Land use conversion cost matrix for each scenario | | | | | | |
|---|---|---|---|---|---|---|---|---|
| | | Cropland | Forest | Shrub | Grass | Water | Barren | Construction |
| 2030_NAS | Cropland | 1 | 1 | 1 | 1 | 1 | 0 | 1 |
| | Forest | 1 | 1 | 1 | 1 | 1 | 0 | 1 |
| | Shrub | 1 | 1 | 1 | 1 | 1 | 0 | 1 |
| | Grass | 1 | 1 | 1 | 1 | 1 | 0 | 1 |
| | Water | 1 | 1 | 1 | 1 | 1 | 0 | 1 |
| | Barren | 1 | 1 | 1 | 1 | 1 | 1 | 1 |
| | Construction | 1 | 1 | 1 | 1 | 1 | 0 | 1 |
| 2030_EPS | Cropland | 1 | 1 | 1 | 1 | 1 | 0 | 0 |
| | Forest | 0 | 1 | 1 | 1 | 0 | 0 | 0 |
| | Shrub | 0 | 1 | 1 | 1 | 0 | 0 | 0 |
| | Grass | 0 | 1 | 1 | 1 | 0 | 0 | 0 |
| | Water | 0 | 1 | 1 | 1 | 1 | 0 | 0 |
| | Barren | 1 | 1 | 1 | 1 | 1 | 1 | 1 |
| | Construction | 0 | 1 | 1 | 1 | 1 | 0 | 1 |
| 2030_UDS | Cropland | 1 | 1 | 0 | 0 | 0 | 0 | 1 |
| | Forest | 1 | 1 | 1 | 1 | 0 | 1 | 1 |
| | Shrub | 1 | 1 | 1 | 1 | 1 | 1 | 1 |
| | Grass | 1 | 1 | 1 | 1 | 1 | 1 | 1 |
| | Water | 1 | 0 | 0 | 0 | 1 | 0 | 1 |
| | Barren | 1 | 1 | 1 | 1 | 1 | 1 | 1 |
| | Construction | 0 | 0 | 0 | 0 | 0 | 0 | 1 |
| 2030_ADS | Cropland | 1 | 1 | 0 | 0 | 0 | 0 | 0 |
| | Forest | 1 | 1 | 1 | 1 | 0 | 0 | 0 |
| | Shrub | 1 | 1 | 1 | 1 | 1 | 0 | 1 |
| | Grass | 1 | 1 | 1 | 1 | 1 | 0 | 1 |
| | Water | 1 | 0 | 0 | 0 | 1 | 0 | 0 |
| | Barren | 1 | 1 | 1 | 1 | 1 | 1 | 1 |
| | Construction | 1 | 1 | 1 | 1 | 1 | 0 | 1 |

surface evaporation, soil water content, canopy interception, and litter water retention. The resulting residual water is considered the water conservation amount for the respective grid cell (*Collignan et al., 2023*). The calculation process requires soil texture, vegetation root depth, precipitation, evaporation and other parameters of each grid unit to estimate water yield. The formula is as follows:

$$Y(x) = \left(1 - \frac{AET(x)}{P(x)}\right) \cdot P(x) \tag{1}$$

$$\frac{AET(x)}{P(x)} = \frac{1 + \omega(x) + R(x)}{1 + \omega(x) \cdot R(x) + \frac{1}{R(x)}} \tag{2}$$

where $Y(x)$ is the annual water volume (mm) of LULC type, $P(x)$ is the grid $x$ annual precipitation (mm), and $AET(x)$ represents the actual evapotranspiration (mm). $\frac{AET(x)}{P(x)}$ is the ratio of actual evapotranspiration to precipitation, and $R(x)$ is defined as the ratio of potential evaporation to precipitation. It has no dimension and can be calculated by formula (3)–(4).

$$\omega(x) = Z \cdot \frac{AWC(x)}{P(x)} \tag{3}$$

$$R(x) = \frac{k(x) \cdot ET_0}{P(x)}. \tag{4}$$

$\omega(x)$ is the ratio of modified vegetation available water to precipitation, without dimension; Z, often referred to as the Zhang coefficient (*Zhang et al., 2004*), is a constant representing the characteristics of seasonal precipitation, with a value range of 1 to 30; $AWC_x$ is the amount of available water (mm) determined by the depth and physical and chemical properties of the soil, which can be calculated by Eq. (5). $k(x)$ is the evapotranspiration function of vegetation, and $ET_0$ is the annual potential evaporation (mm), which is calculated by Eq. (7).

$$AWC_x = MIN\left(SoilDepth_x, RootDepth_x\right) \times PAWC_x \tag{5}$$

$$
\begin{aligned}
PAWC = {} & 54.509 - 0.132 \times SAND - 0.03 \times (SAND)^2 - 0.055 \times SILT - \\
& 0.006 \times (SILT)^2 - 0.738 \times CLAY + 0.007 \times (CLAY)^2 - 2.688 \times OC \\
& + 0.501 \times (OC)^2
\end{aligned}
\tag{6}
$$

$$ET_0 = \frac{0.408\Delta\left(R_n - G\right) + \gamma \frac{900}{T+273} \mu_2 (e_s - e_a)}{0.408\Delta + \gamma\left(1 + 0.42\mu_2\right)} \tag{7}$$

where PAWC is the available water content of vegetation (*He et al., 2022*), $SoilDepth_x$ is the maximum soil depth, and $RootDepth_x$ is the root depth of plant CLAY is the soil clay content (%), SAND is the soil sand content (%), SILT content (%), and OC is the organic matter content (%). $R_n$ is the net radiation MJ/ (m$^2\cdot$ d), G is the soil heat flux, $\gamma$ is the hygrometric constant, $\mu_2$ is the wind speed (m/s), T is the daily temperature $e_s$ is the saturated water vapor pressure (kPa), $e_a$ is the actual water vapor pressure (kPa), $\Delta$ is the slope of the saturated water vapor pressure curve (kPa).

## Water conservation calculation

Based on the calculation results of the water yield module of the InVEST model, the comprehensive consideration of soil permeability and terrain variations among different LULC types profoundly influenced surface runoff. Subsequently, the water conservation quantity and depth of each unit were computed utilizing terrain indices and fundamental soil data. This approach enhances the depiction of the spatial distribution of water

conservation quantity in the basin and elucidates the primary factors influencing it. Please refer to Eqs. (8)–(9) for the calculation methodology.

$$WC = \min\left(1, \frac{249}{V}\right) \times \min\left(1, \frac{0.9D}{3}\right) \times \min\left(1, \frac{K}{300}\right) \times Y \tag{8}$$

$$D = \log 10\left(\frac{DA}{SD \times PS}\right) \tag{9}$$

where: WC is the depth of water conservation, the unit is mm; V is the runoff coefficient, without dimension; D is the topographic index, without dimension; K is soil saturated water conductivity, cm/d; DA is the catchment volume of the basin, and the unit is mm. SD is the soil layer thickness, mm; PS is the percentage slope and has no dimension.

In the calculation process, the runoff coefficients corresponding to various ground features are referred to Table 3.

## Data sources
### Remote sensing image
Initially, DEM data (spatial resolution is 30m) is used to delineate the extent of URHRB. The DEM data were sourced from NASADEM_HGT/001 on the Google Earth Engine (GEE) platform (*Kaur et al., 2023*). Watershed map data were obtained for this study from the Chinese Department of Natural Resources map service website (http://www.tianditu.gov.cn), used to download the standard map reproduction for analysis (plan approval for GS (2019) no. 4345).

GEE is an online remote sensing big data platform jointly developed by Google, the United States Geological Survey and the University of Kentucky (*Hao et al., 2018*; *Sidhu, Pebesma & Camara, 2018*; *Zhao et al., 2021*). The platform includes application program interface file, script manager, resource manager and other modules, which can quickly complete the remote sensing data search, analysis and output, and can realize the rapid visualization of PB-level remote sensing big data (*Kaur et al., 2023*; *Xie et al., 2019*). In this study, remote sensing images from Landsat 5 TM and Landsat 8 OLI, featuring cloud cover of less than 10%, collected from June to September annually, served as data sources, as detailed in Table 4.

### Data related to water conservation
Firstly, based on the previously screened remote sensing images, interpretations were conducted on the GEE platform to obtain the LULC results of the catchment for the years 1990, 2000, 2010, and 2020. Precipitation data for 2030 were compared with climate models published by the Coupled Model Intercomparison Project Phase 6 (CMIP6). Finally, the SSP scenarios (SSP126, SSP245, SSP370 and SSP585) under the BCC-CSM2-MR model newly released by Beijing Climate Center were selected to simulate the precipitation under different development scenarios in 2030, and the water conservation under different development scenarios in 2030 was finally calculated. Additional data sources are detailed in Table 5.

**Table 3  Runoff coefficients corresponding to different landforms and different land use types.**

| Slope | Runoff coefficient (V) | | | | | | |
|---|---|---|---|---|---|---|---|
| | Cropland | Forest | Shrub | Grass | Water | Barren | Construction |
| <10° | 0.20 | 0.03 | 0.04 | 0.05 | 0 | 1 | 1 |
| 10°−20° | 0.23 | 0.03 | 0.06 | 0.11 | 0 | 1 | 1 |
| >20° | 0.26 | 0.04 | 0.07 | 0.12 | 0 | 1 | 1 |

**Table 4  The number of cloud cover less than 10% in Landsat remote sensing images in the study area from 1990 to 2020.**

| Sensors | Number of images/scenes per year | | | |
|---|---|---|---|---|
| | 1990 | 2000 | 2010 | 2020 |
| Landsat5 TM | 154 | 194 | 202 | – |
| landsat7 ETM+ | – | 175 | – | – |
| Landsat8 OLI | – | – | – | 213 |

**Table 5  Water conservation analysis data set.**

| Variable | Format | Spatial resolution | Units | Data sources |
|---|---|---|---|---|
| LULC (1990, 2000, 2010 and 2020) | tif | 30 m | – | Google Earth Engine |
| Precipitation (1990, 2000, 2010 and 2020) | tif | 30 m | mm | https://doi.org/10.5281/zenodo.3185722 |
| Spatial distribution of temperature (1990, 2000, 2010 and 2020) | tif | 30 m | °C | https://doi.org/10.11888/Meteoro.tpdc.270961 |
| Map of ET (1990, 2000, 2010 and 2020) | tif | 30 m | mm | https://doi.org/10.11888/RemoteSen.tpdc.272831 |
| Map of root restricting layer depth | tif | 30 m | mm | HWSD v1.2, http://data.tpdc.ac.cn/zh-hans/data/844010ba-d359-4020-bf76-2b58806f9205/ |
| Map of PAWC | tif | 30 m | % | |
| Maximum root depth for plants in this LULC class | xlsx | – | mm | https://naturalcapitalproject.stanford.edu/ |
| Evapotranspiration coefficient of different LULC class | xlsx | – | Dimensionless | https://naturalcapitalproject.stanford.edu/ |
| Digital elevation model(DEM) | tif | 30 m | m | NASA/NASADEM_HGT/001 https://developers.google.com/earth-engine/datasets/catalog/NASA_NASADEM_HGT_001 |
| Watersheds and Sub-watersheds | shp | – | Dimensionless | ARCGIS 10.6 |

### *LULC simulation and prediction data*

LULC change is the result of multiple driving factors (*Zhang et al., 2022*). Based on the research results of *Liang et al. (2021)* we identified 11 influencing factors, encompassing natural, geographical, and socio-economic aspects. For details, see Table 6. Natural factors decisively influence the mode, intensity, and development direction of land use, primarily through resistance to construction land expansion, indicated by factors such as elevation,

**Table 6  Areas of and changes in each land use type from 1990 to 2020.**

| LULC | Area (km$^2$) | | | | Change Area (km$^2$) | | | |
|---|---|---|---|---|---|---|---|---|
| | 1990 | 2000 | 2010 | 2020 | 1990–2000 | 2000–2010 | 2010–2020 | 1990–2020 |
| Cropland | 17957.11 | 19144.92 | 16990.88 | 14736.76 | 1187.81 | −2154.04 | −2254.12 | −3220.35 |
| Forest | 67494.84 | 68693.09 | 71983.81 | 74647.07 | 1198.25 | 3290.72 | 2663.26 | 7152.23 |
| Shrub | 1565.21 | 763.12 | 389.57 | 150.48 | −802.08 | −373.55 | −239.09 | −1414.72 |
| Grass | 4032.01 | 2420.74 | 1395.16 | 646.62 | −1611.27 | −1025.58 | −748.54 | −3385.39 |
| Water | 777.30 | 656.25 | 758.28 | 1050.15 | −121.05 | 102.03 | 291.87 | 272.85 |
| Barren | 0.91 | 0.41 | 0.03 | 0.40 | −0.50 | −0.38 | 0.37 | −0.51 |
| Construction | 462.80 | 611.64 | 772.45 | 1058.70 | 148.84 | 160.81 | 286.25 | 595.90 |

slope, annual precipitation, annual average temperature, and normalized vegetation index. Geographical factors significantly affect the cost of land use change, primarily through distances from various land uses to urban centers, government institutions, different types of roads, and rivers, which serve as indicators of locational conditions. Socio-economic factors substantially promote the expansion of construction land. Population density and Gross Domestic Product density, key indicators of urban development, were chosen as the driving factors to represent socio-economic influences. All the aforementioned data were sourced from the Resource and Environmental Science Data Platform (https://www.resdc.cn/) and processed using ArcGIS 10.6 software (Product Version: ArcGIS Desktop 10.6.0.8321). The analysis results of Pearson correlation coefficients for the 11 driving factors are depicted in Fig. 3.

# RESULTS

## Spatial–temporal evolution of LULC

The spatial distribution of land use within the URHRB during the years 1990, 2000, 2010, and 2020 predominantly consisted of forest and cropland, with forest land being the most prevalent. Barren land occupied the smallest area, as illustrated in Fig. 4.

For instance, in 2020, the forest and cropland areas measured 74,647.07 km$^2$ and 14,736.76 km$^2$, constituting 79.16% and 15.63% of the URHRB, respectively. The average annual forest coverage in Ningshan, Zhenping, Liuba, and Foping Counties, located at the southern foot of the Qinling Mountains in the URHRB, exceeds 95%. Croplands are primarily located in Hantai District of Hanzhong City, southeast Mianxian County, central Chenggu County, southwest Yangxian County, Hanbin District of Ankang City in the Ankang Basin, and Yunyang and Xichuan Counties in the eastern hilly areas, that terrain is predominantly flat, which corresponds to the extensive area covered by croplands. Construction land is primarily concentrated in and around Hanzhong City in the upper basin, and Ankang, Shangluo, and Shiyan Cities in the middle basin. Additionally, Xixia and Xichuan Counties in Henan Province exhibit high population densities due to their flat terrains. In 2020, the total construction land area spanning 13 counties and districts, including Hantai District in Hanzhong City, was 728.85 km$^2$, representing 68.92% of URHRB's total construction land, as illustrated in Table 6.

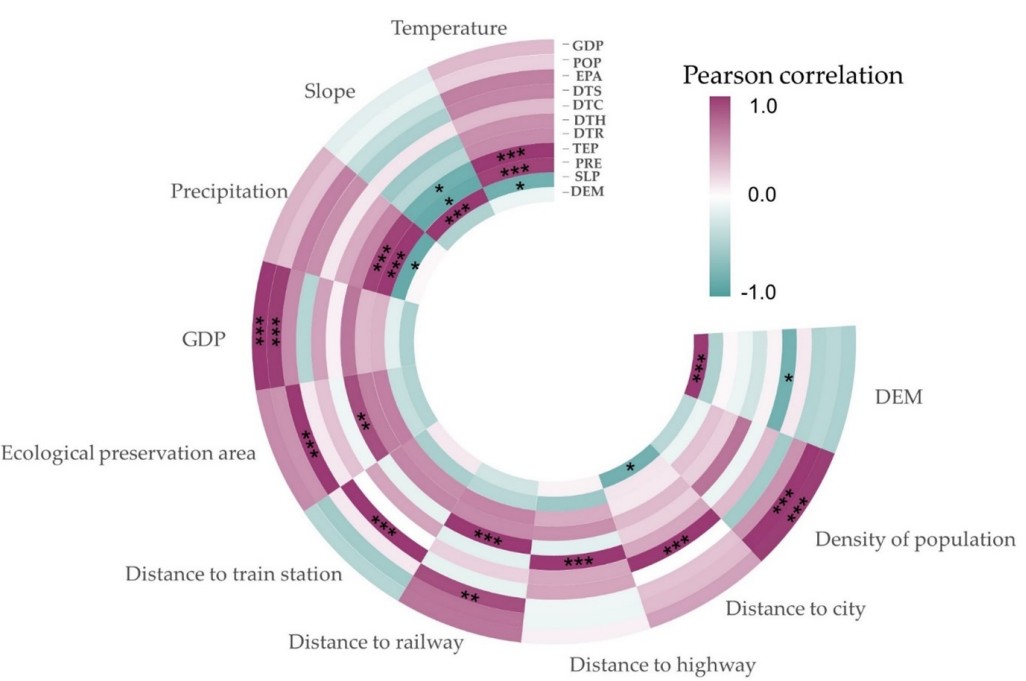

**Figure 3 Pearson correlation coefficients for 11 driving factors to land expansion.**

From the perspective of change trends, the conversion between different land types in the URHRB from 1990 to 2020 is illustrated in Fig. 5. Owing to the "returning farmland to forest" policy initiated in 1999 (*Zhang, Zinda & Li, 2017*), adjustments in the agricultural structure, and the implementation of water source protection measures, the forest area in the water source region has increased by 7,152.23 km$^2$. Influenced by the accelerating urbanization process and the continuous improvement of socio-economic conditions, the construction land area has expanded by 595.90 km$^2$, predominantly concentrated around Hanzhong, Shiyan, Ankang, Shangluo, and other cities. The South-to-North Water Diversion Project has significantly impacted the water area, resulting in an increase of 272.85 km$^2$. Other land use types exhibited a declining trend. Notably, the cropland area initially increased by 1,187.81 km$^2$ from 1990 to 2000, but then began to decrease around 2000, culminating in a total reduction of 3,220.35 km$^2$ by 2020. To sum up, LULC changes of URHRB are mainly shown as significant increase of construction, water and forest, and significant decrease of cropland, shrub and grass, which mainly benefit from the strong implementation of returning farmland to forest and grass in URHRB area. It shows that LULC is affected by severe human factors.

## LULC simulation and prediction

Firstly, the study selected 11 factors including elevation, slope, slope direction, average annual precipitation, average annual temperature, and distance from railway based on the LULC data of 2000 and 2010. The PLUS model was then employed to simulate and predict the land use distribution in 2020 under the natural development scenario. By comparing

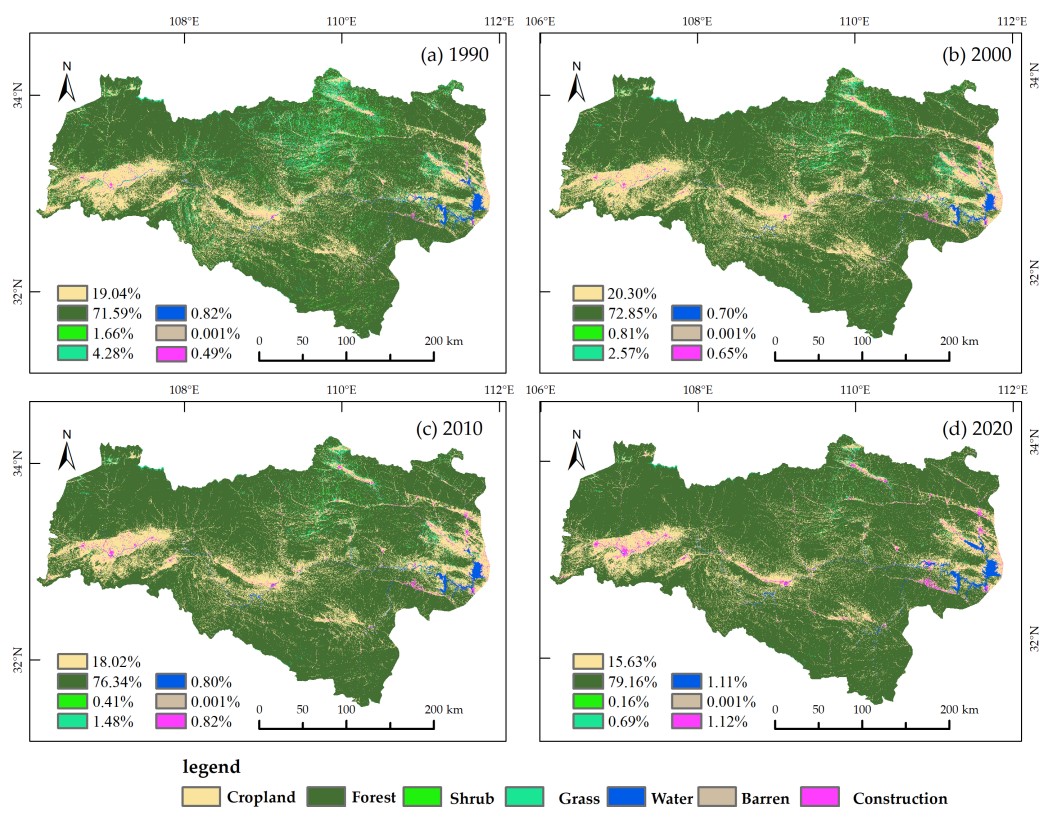

**Figure 4 Results of LULC classification in URHRB, 1990-2020.** Map Source: Tianditu (http://www. tianditu.gov.cn).

and analyzing a 10% random sample with the actual land use in 2020, the Kappa coefficient was calculated to be 0.79, indicating that the PLUS model effectively simulates the land succession process in the study area and accurately predicts land use changes for 2030. The land use simulation results for 2030 in the URHRB reveal significant disparities in land use patterns across different scenarios. Notably, under the environmental protection scenario, the PLUS model even simulates further greening initiatives in comprehensive parks and botanical gardens like the Tianhan Cultural Park and Tianhan Wetland Park in Hanzhong City. These simulation outcomes align with the specific plans for creating a "garden city" in Hanzhong City. Thus, the forecast results offer a deeper understanding of the drivers behind human activities and land use changes in the URHRB, furnishing valuable data and theoretical underpinnings for regional planning. The LULC results under four development scenarios are shown in Fig. 6 and Table 7.

## Simulation results of water conservation

This study first debugs and verifies the model through the annual runoff statistics of China Water Resources Bulletin 2020 and the control stations of important water system nodes in the Yangtze River Basin. The annual runoff of the URHRB is $565 \times 10^8$ m$^3$, and the water yield and water source conservation are calculated by repeated simulation of INVEST model. The Z-coefficient is 27. The water conservation in the URHRB in 2000

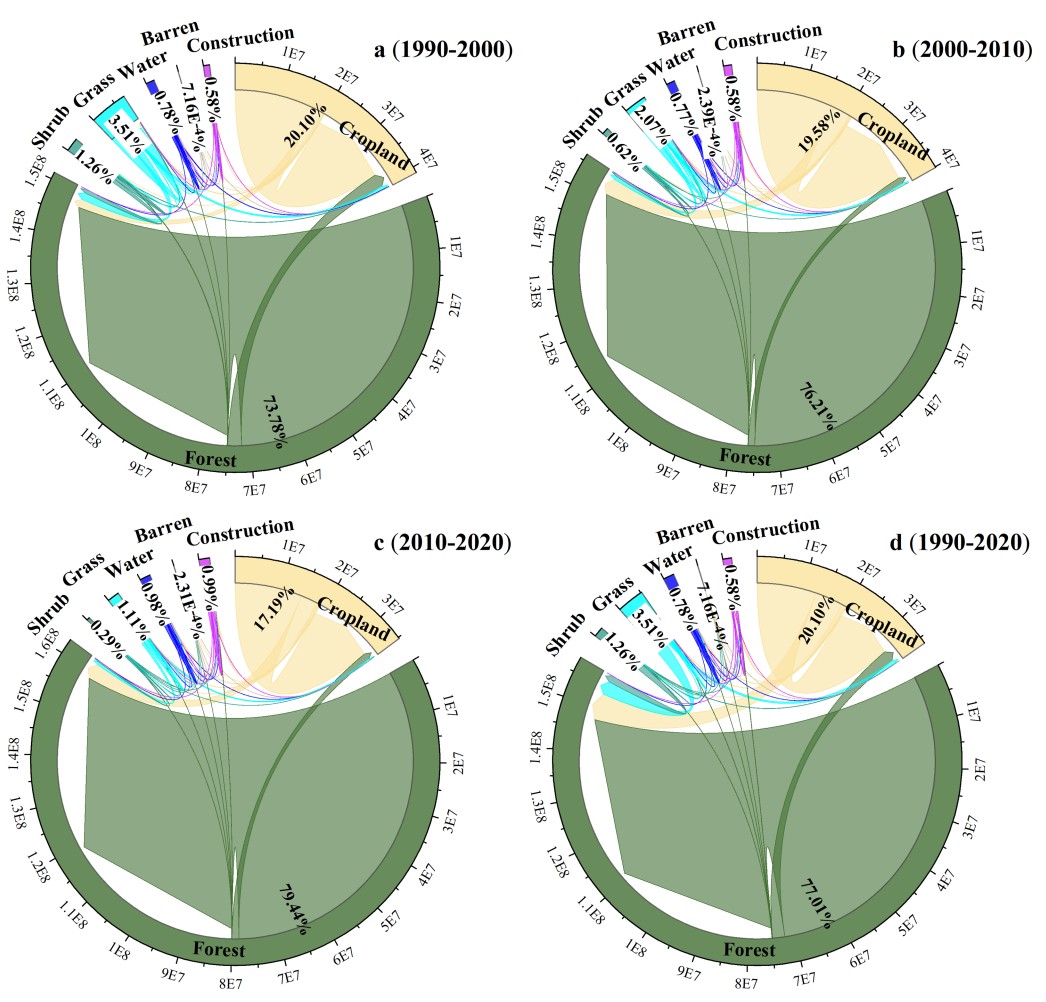

**Figure 5** (A–D) LULC transfer map of annual URHRB between years.

**Table 7  Changes of land use area under four development scenarios in 2030.**

| LULC | Area (km²) | | | | Change Area (km²) | | | |
|---|---|---|---|---|---|---|---|---|
| | NAS | EPS | UDS | ADS | NAS | EPS | UDS | ADS |
| Cropland | 12888.94 | 12526.25 | 14678.64 | 16872.32 | −1847.83 | −2210.51 | −58.12 | 2135.55 |
| Forest | 76369.19 | 76375.60 | 74644.06 | 72504.77 | 1722.12 | 1728.53 | −3.01 | −2142.30 |
| Shrub | 81.15 | 225.73 | 150.23 | 143.86 | −69.33 | 75.24 | −0.26 | −6.62 |
| Grass | 352.51 | 969.93 | 641.24 | 627.22 | −294.10 | 323.31 | −5.38 | −19.40 |
| Water | 1317.83 | 1155.17 | 1050.68 | 1104.11 | 267.68 | 105.02 | 0.53 | 53.96 |
| Barren | 0.33 | 0.01 | 0.35 | 0.38 | −0.06 | −0.39 | −0.05 | −0.02 |
| Construction | 1280.23 | 1037.51 | 1124.98 | 1037.52 | 221.53 | −21.19 | 66.28 | −21.17 |

is $570.20 \times 10^8$ m³, and the error is 5.57%, which shows that the model simulation results have good robustness. According to the spatial distribution of water conservation in the URHRB from 1990 to 2020 simulated by InVEST water conservation model (Fig. 7).

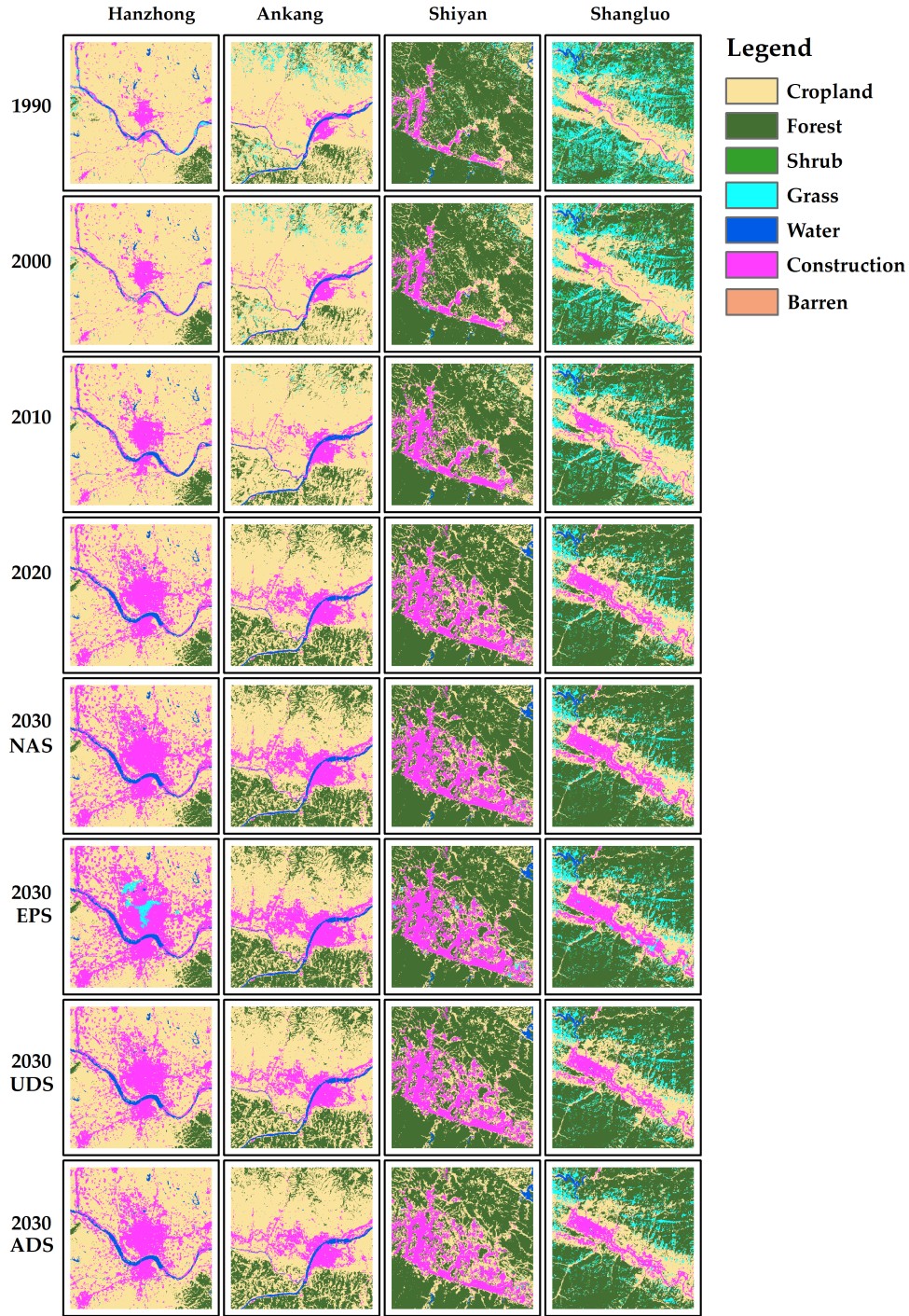

**Figure 6** **Change map of different land use types in four typical cities in the URHRB in 2030 under four development scenarios.** Map source: GEE (https://code.earthengine.google.com/a391359218d63d137710cb4f3dad0131).

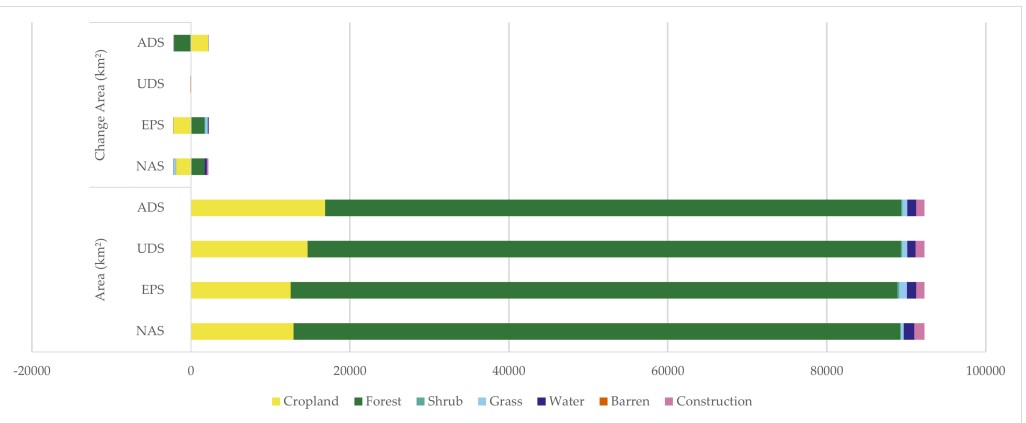

**Figure 7  Area changes in various land use types under four development scenarios in 2030.**

**Table 8  Analysis of water conservation trends from 1900 to 2020, alongside projections under four distinct scenarios for 2030.**

| Annual / 2030 forecast scenarios | Simulation results of water conservation | | | | |
|---|---|---|---|---|---|
| | Mean precipitation (mm) | Mean potential evapotranspiration (mm) | Mean actual evapotranspiration (mm) | Mean water yield (mm) | Total volume ($10^8 m^3$) |
| 1990 | 917.80 | 1055.88 | 298.36 | 619.13 | 571.39 |
| 2000 | 1033.46 | 1055.55 | 312.20 | 721.04 | 665.45 |
| 2010 | 1062.73 | 1038.99 | 314.31 | 748.17 | 690.49 |
| 2020 | 916.24 | 1038.40 | 298.24 | 617.83 | 570.20 |
| 2030_NAS | 890.78 | 1034.38 | 292.51 | 598.18 | 552.06 |
| 2030_EPS | 902.56 | 1032.50 | 297.41 | 622.95 | 574.92 |
| 2030_UDS | 901.79 | 1037.69 | 293.71 | 607.87 | 561.00 |
| 2030_ADS | 799.44 | 1040.92 | 281.89 | 517.32 | 477.43 |

### Temporal and spatial changes of water conservation in URHRB during 1990-2020

Based on the simulation results of water yield and water conservation formula, the water conservation quantity of the URHRB from 1990 to 2020 was obtained. As shown in Table 8, from the perspective of time, there is a big difference in the value of water conservation in the URHRB, and the water conservation in the URHRB presents a trend of first increasing and then decreasing. In 1990, the water conservation volume was $571.39 \times 10^8$ m$^3$, and reached a maximum value of $690.49 \times 10^8$ m$^3$ in 2010. In 2020, it will further drop to $570.20 \times 10^8$ m$^3$. Compared with 1990, it decreased by $1.19 \times 10^8$ m$^3$.

There are obvious differences in the distribution of water conservation in the URHRB, and there are certain rules. For example, through the spatial pattern analysis of four years, it is found that the region with high water conservation is mainly concentrated in the western and southern regions, while the region with low water conservation is mainly concentrated in the northeast, and gradually decreases from west to east. There are common high and
low value regions. During 1990, Taibai County, Liuba County, Foping County and other counties at the southern foot of Qinling Mountain showed obvious water conservation capacity. The areas with medium water conservation are mainly distributed in Xunyang City, Baihe County, Xixia County, Shangnan County and Danjiangkou City in the middle of the URHRB. However, in the upper reaches of the Han River in Hanzhong City, including Mianxian County, Chenggu County, Yang County, Shangluo City, Danfeng County and other densely populated areas, water conservation is obviously low. The main reason should be that the contradiction between people and land in densely populated areas is prominent, and a large number of cropland causes deforestation and serious vegetation destruction, thus causing serious soil erosion. The above regions also have a common feature of relatively low annual precipitation. With the increase of elevation, the water conservation in the basin increases first and then decreases. The low value of water conservation is mainly in the area above 3,000 m, and the high value area is mainly distributed between 200 m and 2500 m. On the slope, the water conservation in the basin showed a gradual increasing trend with the increase of the slope, and the increasing trend was relatively gentle. According to the spatial distribution of land use types in the study area (Fig. 4), land use types such as forest land, cropland, shrubs and construction land have relatively high water conservation, while water and bare land have relatively low water conservation. It can be seen that due to the high forest cover in the region, URHRB's better water conservation function is the main reason why it can provide continuous water supply for the water transfer project of the Middle Route of the SNWD.

## Spatial–temporal changes of water conservation in URHRB under different scenarios in 2030

Through simulation, we obtained the 2030 water conservation of URHRB under different development scenarios, and the results were shown in Fig. 8. Overall, there is a great difference between the URHRB water conservation in 2030 and that in the past 30 years, with the average annual water conservation depth decreasing from the peak value of 748.17 mm in 2000 to 586.58 mm, a relative decrease of 21.58%. The total water conservation decreased from the highest value of $690.49 \times 10^8$ m$^3$ in 2010 to the average value of $541.36 \times 10^8$ m$^3$ under the four development scenarios in 2030, a relative decline of 21.60%. By 2030, water conservation in the EPS scenario is the highest, reaching $574.92 \times 108$ m$^3$, which reveals the development direction of URHRB in the future. Therefore, in order to achieve sustainable and reliable water supply of URHRB, people need to continue to carry out ecological environment construction in the future. The results of this study will provide some reference for the water transfer project of the Middle Route of the SNWD.

## DISCUSSION

### Impact of LULC change on water conservation

In general, water conservation function within a basin is influenced by regional climate, basin size, topography (slope), land cover characteristics (such as soil texture and depth), and human activities. Long-term research results show that the impact of land use change on water conservation is a complex process, land use change will affect the underlying surface

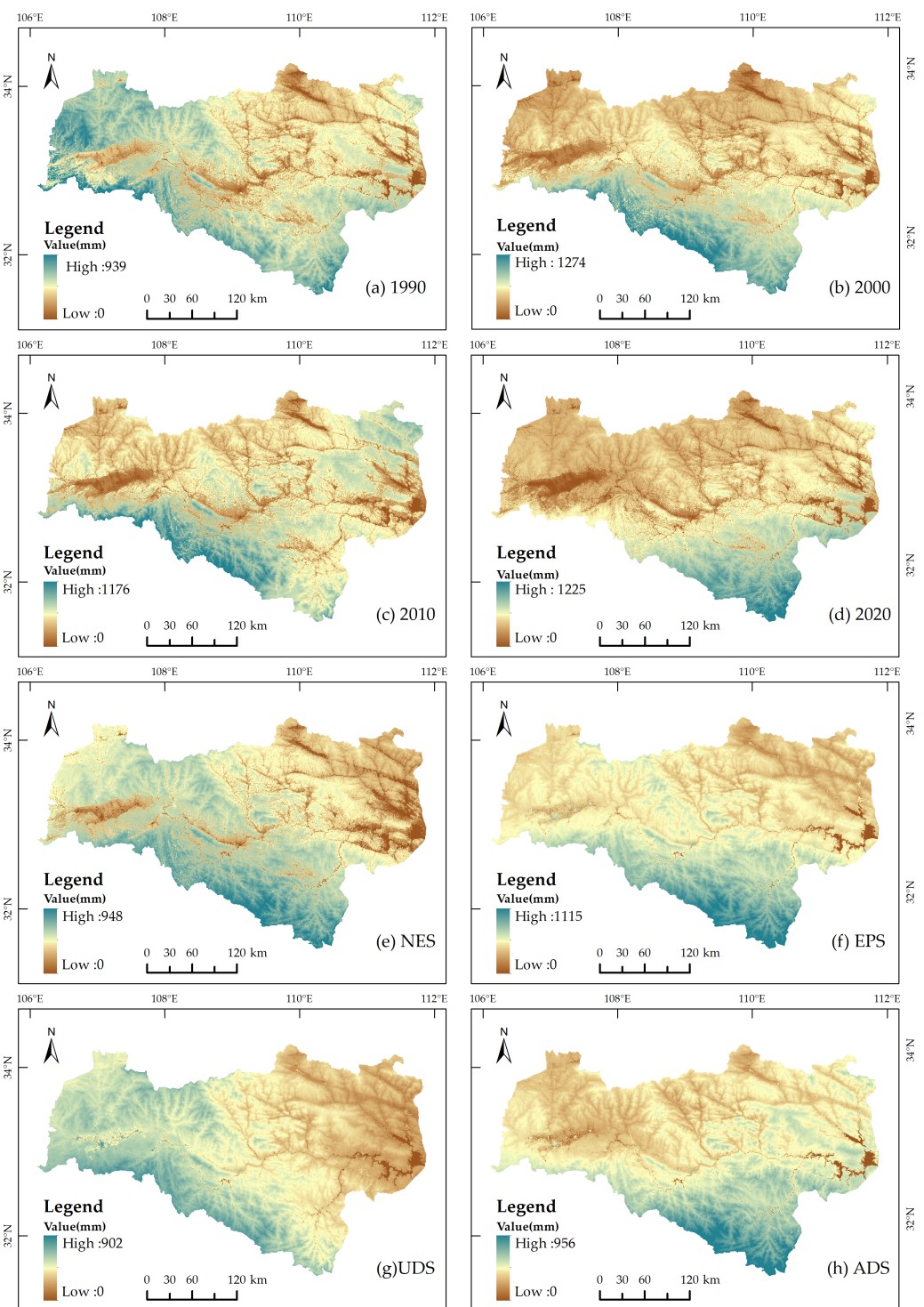

**Figure 8   Water conservation in URHRB from 1990 to 2020 and forecast map of water conservation in URHRB in 2030.** Map source: Tianditu (http://www.tianditu.gov.cn).

type and structure, and then affect the water conservation function. Based on InVEST model, this study presents the land use change and water retention function of URHRB in a quantitative and visual way, and quantitatively analyzes the impact of regional land use change on water retention function. The results show that the forest water conservation function of ecological land is higher, mainly because the forest vegetation intercepts precipitation through the canopy and absorbs precipitation through the litter layer. Its deep root system further promotes the infiltration of precipitation into the soil layer, thus reducing surface runoff and showing high water conservation ability (*Cantillo Polo, 2020*; *Chandler et al., 2018*). At the same time, the soil under the tree can not only store rainwater, but also reduce the peak flow, increase the dry season flow, and improve the utilization efficiency of water resources. The water conservation capacity of farmland, shrubland and grassland was similar to that of forest, and the water conservation function was ordered as follows: forest >grassland >shrubland >water area. When the area of each land type increased by 1km$^2$, the water conservation increased by $65.39 \times 10^4$ m$^3$, $59.33 \times 10^4$ m$^3$, $59.03 \times 10^4$ m$^3$ and $0.18 \times 10^4$ m$^3$, respectively. Refer to Fig. 9 for the results of water conservation under different conditions of various types of LULC. The above conclusions are consistent with the results of the Wei River study (*Wang et al., 2024*). Overall, from 1990 to 2010, the proportion of forest land with a greater impact on water conservation increased from 73.13% to 80.88%, and the proportion of construction land increased from 0.50% to 1.15%, both showing an increasing trend, so the amount of water conservation also gradually increased. Among them, the total water conservation of forest increased from $425.98 \times 10^8$ m$^3$ in 1990 to $503.52 \times 10^8$ m$^3$ in 2000, and reached $546.85 \times 10^8$ m$^3$ in 2010, which is the most important water conservation unit in URHRB.

**Impact of climate change on water conservation**

It is well known that climate change affects water yield primarily through changes in precipitation and actual evapotranspiration (*López-Moreno et al., 2011*). At the same time, some scholars found in the study of watershed water yield in the source area of the Three Rivers in China that land use change would cause local water vapor cycle, and then affect surface runoff change (*Guzha et al., 2018*), and will affect regional water conservation by changing regional climate.

The regions with high water conservation and high precipitation in the water source region maintained a high spatial consistency, and both showed a rapid decline trend from south to north. Among them, the average precipitation and water conservation depth in the Daba Mountains and the western Qinling Mountains are relatively high, which is because the water vapor transport brought by the southwest air flow and the south air flow converge in the Qinba Mountains, and the water vapor transport and the southwest air flow brought by the south jet stream converge in the northern foot of the Daba Mountains and rise along the windward slope of the Qinling Mountains and the Daba Mountains (*Fang et al., 2002*; *Wang et al., 2021*), resulting in relatively strong precipitation. Other studies show that the impingement of the Three Gorges Reservoir also indirectly leads to the increase of precipitation in the Daba Mountains and Qinling Mountains. From 1990 to 2020, the average annual precipitation in the URHRB was 917.80 mm, 1,033.46 mm, 1,062.73 mm

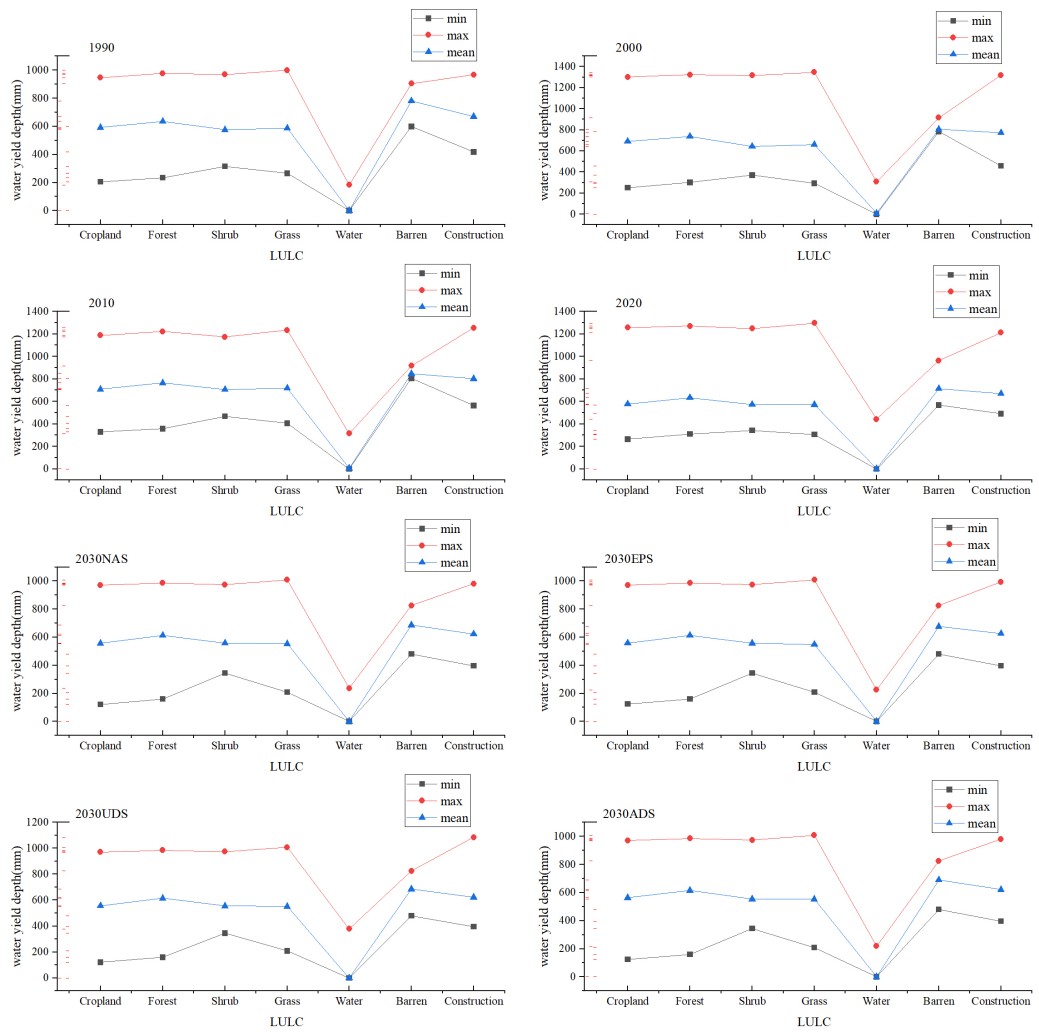

**Figure 9** Statistical analysis of water conservation across various land types in different years and under distinct development scenarios.

and 916.24 mm, respectively, showing a trend of first increasing and then decreasing, which was the same with the trend of water conservation in the URHRB.

## Analysis of changes in water conservation under four development scenarios in 2030

The research area of this paper is an important water source in China. The analysis of the water retention capacity of the region and the scientific prediction of the future water retention are of reference value for further improving the ecological protection measures of URHRB, maintaining the stability of the ecosystem, improving the carrying capacity of water resources, and are also of great significance for maintaining the continuous water transmission of livelihood projects such as the SNWD.

First of all, in the process of natural development scenario simulation, considering the continuous growth of population, the construction land will increase at the current

development rate. Under the influence of the current ecological protection policy, the water conservation function of URHRB has been further consolidated, and the water area in 2020 will be 1,050.15 km$^2$, accounting for 1.14% of the total area of URHRB. In this scenario, it will increase to 1,317.83 km$^2$ by 2030, accounting for 1.43% of the total URHRB area. The simulation process of ecological protection scenario is close to the natural development scenario, but the construction land development is strictly controlled, and the expansion area is reduced by 242.72 km$^2$ compared with the natural development scenario. Under the construction and development scenario and the cropland protection scenario, the cropland area will increase compared with 2020, reaching 14,678.64 km$^2$ and 16,872.32 km$^2$ respectively, while the forest area will decrease to 74,644.06 km$^2$ and 7,2504.77 km$^2$ respectively. Based on the above analysis, the water conservation function of water source mainly depends on the land characteristics such as forest land and cropland. Therefore, under the NAS, EPS, UDS and ADS, the total water conservation of URHRB in 2030 is $552.06 \times 10^8$ m$^3$, $574.92 \times 10^8$ m$^3$, $561.00 \times 10^8$ m$^3$ and $477.43 \times 10^8$ m$^3$ respectively. From what has been discussed above, we can conclude that in the case of environmental protection, the total amount of water saving is slightly higher, and the water source land use planning is also the most reasonable. This is similar to the results of water conservation research in Shaanxi Province (*Liu et al., 2018*).

## URHRB development planning based on water conservation function protection

Based on the above research results, we believe that URHRB should pay full attention to the protection of ecological environment, strengthen the monitoring and protection of ecological land such as forests, grasslands, shrublands and water areas, effectively improve the capacity of water resources conservation, and ensure the health and stability of regional ecosystems. In addition, we also propose to divide the core security area, water conservation control area and water conservation ecological protection area in the URHRB. The water area of Danjiangkou Reservoir and its surroundings up to Baihe County in Shaanxi province is the core protection area, the Duhe River basin and the Hanjiang River basin above Baihe County up to Ankang urban area are set as water conservation control areas, and the other areas are water conservation ecological protection areas. In the core security area, attention should be paid to the management of sloping farmland around Danjiangkou Reservoir and the construction of clean small watershed, so as to provide guarantee for regional water conservation and water purification. In the water conservation control area, we should continue to do a good job in monitoring urban and rural land expansion and comprehensive environmental management, slow down the expansion rate of cultivated land and cities, and further maintain and improve the current scale and pattern of ecological land. Water conservation ecological protection areas can focus on ecological protection and water conservation, and gradually carry out ecological protection and comprehensive management. The above measures ensure that the future water conservation function of URHRB is fully protected (*Li et al., 2021a*; *Wang et al., 2022*; *Wittwer et al., 2021*).

**Research limitations**

In this study, the GEE platform was used to classify the LULC of URHRB from 1990 to 2020. Then, the PLUS model was used to simulate and predict the land use of URHRB in 2030 under four development scenarios. In general, the InVEST-PLUS model has a good simulation effect in the study, effectively quantifying and simulating the overall change law of URHRB water conservation in the past 30 years and in 2030 (*Liu, Zhang & Lin, 2023*). It is helpful to deepen human understanding of the function of water resources production and water resources protection. However, because the InVEST model only combines the empirical data of existing achievements with specific parameters such as evapotranspiration coefficient, maximum root depth and Z coefficient, and does not consider the soil moisture content or the precipitation retention of different vegetation under different slopes, the calculated water quantity inevitably has certain errors (*Li et al., 2021a*; *Yang et al., 2019*). At the same time, when calculating the annual water quantity and water source conservation in 2030, the annual precipitation data and evapotranspiration data used are both climate prediction products, which have certain uncertainties. In addition, there are some errors in the prediction accuracy of the CA-Markov model (*Mondal et al., 2013*; *Wang, Stephenson & Qu, 2019*; *Wu et al., 2019*). Further research and improvement are needed. In the future, we will consider optimizing and improving other models, and then further validation of the current study to improve the accuracy of URHRB's water conservation prediction.

# CONCLUSIONS

The Middle Route of the SNDW is a significant strategic initiative to allocate water resources across basins and regions in China. Since the project began to transfer water, it has greatly alleviated the severe shortage of water resources in North China. Consequently, safeguarding the water conservation function of the water source entails significant political, economic, social, and ecological benefits. This study delineates the spatial–temporal characteristics of water conservation changes within URHRB over the past 30 years, examines the mechanisms by which land use changes affect water conservation function, and forecasts water conservation under various development scenarios for 2030 in URHRB. Finally, the study draws the following conclusions:

1. The water conservation in the URHRB exhibited spatial consistency, primarily concentrated in the southern Daba Mountain area and the western middle-altitude mountain area, with an overall increasing trend from west to east and north to south. Simultaneously, forest land and cropland within URHRB serve as the primary contributors to water conservation. Between 1990 and 2020, the proportion of water conservation provided by forest land within the total catchment area will increase from 75.05% to 82.99%. Conversely, the corresponding proportion of cropland decreased from 19.69% in 1990 to 14.97% in 2020. Together, water conservation from these two sources accounts for over 93.74% of the total water catchment area. Following these are grassland, shrubland, construction land, and unused land. Hence, the initiative to convert farmland back to forest and grassland is conducive to enhancing water conservation in URHRB.

2. From 1990 to 2020, the total water conservation in URHRB exhibited an initial increase followed by a decrease, with average water conservation values ranging from

$570.20 \times 10^8$ m$^3$ to $690.49 \times 10^8$ m$^3$, peaking in 2010. Prediction results for 2030 indicate that under four scenarios, the ecological protection scenario—which prioritizes ecological land preservation and restricts construction land development—yields the highest water conservation, estimated at $552.22 \times 10^8$ m$^3$. However, overall, the forecast for 2030 reveals a significant downward trend in water conservation within URHRB, necessitating urgent attention from local management authorities to devise effective countermeasures, ensuring the sustained water supply from the Danjiangkou Reservoir.

3.   To maintain high-quality water supply services, URHRB should implement zoning and classified management measures based on quantified water conservation assessments. This includes enhancing ecological land management in Hanzhong City, the southern region of Ankang City, and the northern Qinling Mountain area, as well as undertaking initiatives such as natural forest protection and restoration, wetland ecological restoration, and key protection forest construction projects within URHRB. These efforts are aimed at enhancing the water resource yield capacity to safeguard the health and stability of the environmental ecosystem within URHRB.

## ACKNOWLEDGEMENTS

The authors would like to thank the State Key Laboratory of Geohazard Prevention and Geoenvironment Protection (Chengdu University of Technology) for providing assistance during the experimental process. We sincerely thank the editor, associate editor, and anonymous reviewers for critical evaluation and constructive suggestions to improve the manuscript.

### Funding

This research was funded by Henan Province 2023 Science and Technology Development Plan Science and Technology Research Project (No:23210232027). The funders had no role in study design, data collection and analysis, decision to publish, or preparation of the manuscript.

### Grant Disclosures

The following grant information was disclosed by the authors:
Henan Province 2023 Science and Technology Development Plan Science and Technology Research Project: No: 23210232027.

### Competing Interests

Yi Cao is employed by Sinopec Northwest China Petroleum Bureau.

### Author Contributions

- Pengtao Niu conceived and designed the experiments, performed the experiments, analyzed the data, prepared figures and/or tables, and approved the final draft.
- Zhan Wang analyzed the data, authored or reviewed drafts of the article, and approved the final draft.

- Jing Wang performed the experiments, authored or reviewed drafts of the article, and approved the final draft.
- Yi Cao analyzed the data, prepared figures and/or tables, and approved the final draft.
- Peihao Peng conceived and designed the experiments, authored or reviewed drafts of the article, and approved the final draft.

## Data Availability

The LULC code links and all datasets and raw measurements are available in the Supplementary Files.

## Supplemental Information

Supplemental information for this article can be found online at http://dx.doi.org/10.7717/peerj.18441#supplemental-information.

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
