# Peer review of "Estimation and prediction of water conservation in the upper reaches of the Hanjiang River Basin based on InVEST-PLUS model"

_PeerJ, doi:10.7717/peerj.18441_

## Round 0.1 · original submission · Major Revisions

Reviewers have identified major weakness in novelty, methods, and publish ability. I am requesting a major revision following each comment of the reviewers. Particular attention should be to do a major revision that will produce novelty in methods and clarity in writing.

Reviewer 1 ·

Basic reporting

This manuscript used the InVEST-PLUS model to provide a detailed estimate and prediction of water-holding in the Upper Han River Basin. The authors combined soil permeability and topographic variations to calculate the amount and depth of water reclamation in each unit, and enhanced the description of the spatial distribution of water reclamation using topographic indices and basic soil data. While the selected topics are of some practical significance, but the research methodology and research content used in the selected topics are less innovative than existing research in the field. Similar to the article "Estimation and prediction of water conservation in Shaanxi Province based on the InVEST-PLUS model" published in the Journal of Soil and Water Conservation by Zhou Pingping et al, a conventional model was used for the analysis, and there is no model improvement or theoretical innovation in this manuscript. In addition, there are problems and deficiencies in the introduction, data, methodology, results, discussion, etc., and a large number of FLUS models appear in the paper, while the title of the manuscript is " Estimation and prediction of water conservation in the upper reaches of Hanjiang River Basin based on InVEST-PLUS model", and for mentioning the need to use the FLUS model, please clarify whether you are using the FLUS model or the PLUS model.The quality of your manuscript needs to be substantially improved and is currently not up to The quality of your manuscript needs to be greatly improved and is not up to publication standards at this time. Specific comments are set out below:

Experimental design

(1) L127-236 lists the research methodology, but does not highlight the advantages of the chosen methodology over other methods or demonstrate the superiority of the chosen modeling approach. In addition, the introduction lacks a comprehensive summary of previous research efforts.
(2) The introductory section does not explain what specific problem this paper addresses and make it clear what the innovation of this paper is in comparison to other articles of the same type.
(3) The introductory section lacks a comprehensive summary of previous research, as well as a description of how this paper goes further in previous research.
(4) In L104-L107 " The FLUS model can integrally incorporate the rule framework based on the Land Expansion Analysis Strategy (LEAS) with the CA model to achieve de-tailed simulations of future LULC changes (Liang et al. 2018; Liu et al. 2017)." This is supposed to be a description of the PLUS model, why is there "FLUS model", please explain.
(5) L107-L110 have been reviewing the "FLUS model" without mentioning the "PLUS model" that you used, which is really puzzling. Suggest checking the full text.
(6) There is no mention in L49-L114 of why the PLUS model was chosen.
(7) The relevant introduction to the PLUS model in the introduction is suggested to be supplemented.
(8) Four scenarios are mentioned in L134-L135, why are only these four scenarios considered and not the others? What are the criteria for selecting these particular scenarios?

Validity of the findings

(9) Four scenarios are mentioned in L134-L135, and while I, and most potential readers of this paper, can understand the UDS scenarios (Urban Development Scenarios), I would like to see more elaboration on the four scenarios.
(10) The spatial resolution of 30 metres used, as mentioned in L145, may not be sufficient to capture some of the subtle topographic variations, which may introduce a certain degree of error in the precise assessment of water containment.
(11) The InVEST-PLUS model, although widely used for ecosystem service assessment, does not differentiate in detail between surface water, groundwater and baseflow in the context of water harvesting, which may result in the accuracy of the results being compromised.
(12) The research methodology of this paper mainly relies on the InVEST-PLUS model, without introducing other models or methods for comparative analyses, with a lack of comparison of results and accuracy.
(13) The use of the InVEST-PLUS model in ecosystem services assessment is already widespread, and this paper does not demonstrate unique methodological innovations.
(14) In L257-L256, it is mentioned that 11 influencing factors were chosen, while there are many influencing factors, why these influencing factors were chosen.
(15) In L249-L251, it is mentioned that the monthly Landsat 5 TM and Landsat 8 OLI are selected for June-September of each year, please explain why the images are selected for that time period.
(16) In Figure 3, the colours in "Forest, Shrub, Grass" are too close to each other, making it difficult to distinguish between them, and it is recommended that the figure be redone.
(17) The area share of the various classifications in Figure 3 suggests that the sub-maps be accompanied by a corresponding area share map in order to improve the readability of the maps.
(18) It is mentioned in the text that the results of the study will provide some reference for the South-to-North Water Diversion Central Transfer Project, indicating that the policy has a greater impact on the region, why is the policy impact not taken into account in the scenarios modelled in the future?
(19) What is the significance of the modelling of these four scenarios proposed in the paper when the high and low values of NES, EPS, UDS, and ADS are exactly the same in Fig. 8, and the four plots are too similar to be clearly differentiated?
(20) Figure 9 is too poorly readable and it is suggested that other types of diagrams or better representations be modified to improve the readability of the visualisation.
(21) In the discussion section, the article does not make relevant recommendations based on the findings of the study.
(22) Four scenarios are presented in the paper, but the discussion does not address the different protection strategies and measures for the four scenarios.
(23) The FLUS model is again mentioned in L478-L481, and it is recommended to double-check whether the FLUS model or the PLUS model is used in the paper.
(24) "Investment-FLUS model" appears in L479, which is inconsistent with "InVEST-PLUS" in the preceding text, so please check for similar inconsistencies in descriptions throughout the text.
(25) Please create a discussion around your findings.
(26) The study is based on research conducted in the Upper Han River Basin, and your research is specific to the region, and whether it is relevant and generalisable to other regions and globally.
(27) The article does not mention future research directions, please provide future research directions.
(28) There are grammatical problems throughout the text, for example, "the" should be added in front of "Hanjiang River" in the title, so please double-check the text for grammatical problems.
Overall, the manuscript lacks innovation, there are many problems in content and grammar, many FLUS models appear in the text, the manuscript is of low quality, please correct your attitude. Please re-examine the document carefully and improve it. It is recommended that the authors make major revisions or rewrites.

Reviewer 2 ·

Basic reporting

1、The manuscript applies the InVEST model to evaluate the spatial and temporal changes in water conservation of the upper reaches of Hanjiang River Basin over the past 30 years, and combines it with the PLUS model to predict water conservation under different scenarios in the future. Although the manuscript is somewhat innovative, it is still quite far from publication and needs major revision.
2、The Abstract requires major revision. Line 37-38 uses “water production”, while Line 40-41 uses “water conservation”. Water conservation and water production are not equivalent. Water conservation is the subject of this paper and should be highlighted.
3、In the Discussion section, strategy should be proposed at the right time to show the research significance of this paper.
4、The manuscript contains many mistakes. Please recheck and revise.
5、Line 62 is missing a “,”, Line 311 has an extra “.”, Line 490 has an extra “.”, Line 392 has an extra space, and Line289 has an extra space, so please check throughout!
6、Line 211, the title should be amended to read “Calculation of water conservation”, with reference to the above.
7、Line 262, 266, NDVI and GDP appear only once and can be used in full.
8、Line 269, “ARCGIS” should be changed to “ArcGIS”.
9、Line 312-319 should be described in the “research methods”.
10、“LULC simulation and prediction” needs to be redescribed, too much of it belongs in the methods section rather than the description of the results in this paper.
11、Many descriptions of the Results appear in the Discussion section.
12、Line 454, “South-to-North Water diversion project” should be replaced by “SNWD”.
13、Line 27-28, please re-describe.
14、Line 31, “cultivated land, scrub and grassland” is completely inconsistent with the later description of land use types.
15、Line 39, Why is the “Three Gorges Reservoir area” specifically described?
16、Line 42, What does “visual analysis” refer to?
17、PLUS and FLUS use confusion in this article. FLUS appears in the Keywords section. This is an extremely serious mistake.
18、Line 148, What does “ee.simple Composite () algorithm” refer to?
19、Line 104, Does FLUS include LEAS?
20、Line 108, Please check whether the description “Liu Tao. et al” is appropriate.
21、Figure 4 should be depicted in more detail.
22、Please standardize some descriptions, e.g. “Spatiotemporal” and “Spatial-temporal”, “water resource conservation”, “water conservation”, and “water conservation functions”.

Experimental design

1、Line 488, What are the data sources for climate prediction products?
2、Why do we need to produce LULC data, and what are the differences and necessities compared with the existing data?
3、What is the basis for future scenario design?
4、The setup parameters for the four scenarios in the PLUS model should be given.

Validity of the findings

1、What is the Kappa coefficient for the PLUS model, 0.7 at Line 308, 0.79 at Line 174?
2、Line 443, “precipitation is the main influencing factor ……”, The paper does not quantitatively analyse the influencing factors, so such conclusions should be described with caution.
3、Line437, why there are four averages in 1990-2000.
4、The setup parameters for the InVEST model and the PLUS model could perhaps be given in the supplementary material.

---

## Round 0.2 · Minor Revisions

There are still improvements needed in the writeup.

Reviewer 1 ·

Basic reporting

Although the author has made some revisions to the introduction, results, discussion, and figures of the article, there are still issues in these sections. For example, the results section contains significant errors in some of the data descriptions, and the textual descriptions do not correspond with the data presented in the figures. Therefore, the author needs to further improve the obvious errors to meet the standards of this journal. Before the manuscript can be recommended for publication, the author is requested to carefully review the article and make necessary revisions. Below, I have outlined the issues that the author should address:
1.Although the author has made some adjustments to the introduction, the innovative aspects compared to similar studies are still not sufficiently highlighted.

Experimental design

I think the experimental part has been revised properly.

Validity of the findings

1.In the results section, for example, Lines 316-317 mention “For instance, in 2020, the forest and cropland areas measured 74,647.07 km² and 14,736.76 km², constituting 80.88% and 15.97% of the URHRB, respectively.” However, these numbers do not correspond with the data shown in Figure 4. Please carefully check for such types of erroneous descriptions.
2.The results section needs more qualitative descriptions to complement the quantitative statistical descriptions.
3.The discussion section could benefit from additional evidence to enhance the persuasiveness of your argument.

---

## Round 0.3 · accepted · Accept

The authors have responded to the reviewer's comments, and the manuscript is ready for publication.